# Performance Optimization in Streetlifting: The Combined Role of Nutrition, Lean Mass, and Sleep

**DOI:** 10.3390/nu18010105

**Published:** 2025-12-28

**Authors:** Cristian Stranieri, Alessandra Bulbarelli, Elena Lonati, Paola Palestini, Emanuela Cazzaniga

**Affiliations:** 1School of Medicine and Surgery, University of Milano-Bicocca, via Cadore 48, 20900 Monza, Italy; cristian.stranieri2001@gmail.com (C.S.); alessandra.bulbarelli@unimib.it (A.B.); elena.lonati1@unimib.it (E.L.); paola.palestini@unimib.it (P.P.); 2Master’s Programme in Applied Food and Nutritional Sciences, School of Medicine and Surgery, University of Milano-Bicocca, 20900 Monza, Italy

**Keywords:** streetlifting, calisthenics, strength athletes, body composition, nutrient timing, protein intake, creatine, sleep, recovery

## Abstract

Background and Objective: Streetlifting is a developing strength sport derived from calisthenics and based on maximal external load performance in weighted pull-ups, dips, muscle-ups, and squat variations. Its rapid global expansion has raised interest in identifying sport-specific nutritional and recovery strategies that can support performance and health. However, scientific evidence directly focused on streetlifting remains limited. This narrative review aims to summarize current knowledge regarding body composition, nutrition, supplementation, and sleep in streetlifting athletes by integrating findings from related strength sports. Methods: A narrative review design was adopted due to the scarcity of empirical studies on streetlifting. Searches were performed using the terms “streetlifting AND nutrition,” “streetlifting AND body composition,” and “streetlifting AND sleep quality.” Peer-reviewed studies involving comparable strength disciplines were included when directly applicable to performance or recovery determinants. Results: Performance in streetlifting appears strongly driven by strength-to-bodyweight ratio, supported by low-to-moderate fat mass and adequate lean mass. Evidence from resistance training literature suggests that meeting energy requirements, consuming 1.2–1.5 g/kg/day of protein, and using nutrient timing around training may enhance muscle protein synthesis and glycogen replenishment. Creatine supplementation shows consistent benefits for maximal strength and ATP turnover, whereas other supplements lack robust evidence. Sleep duration and quality contribute to neuromuscular recovery, endocrine balance, and cognitive readiness, though sport-specific findings are insufficient. Conclusions: Streetlifting athletes may benefit from integrated nutritional planning, evidence-based supplementation, and sleep optimization. Further sport-specific interventional and longitudinal studies are required to develop validated performance and health guidelines.

## 1. Introduction

Calisthenics is a worldwide training methodology based on bodyweight exercises [1]. According to information from the American College of Sports Medicine, calisthenics were identified as the seventh most popular activity trend in 2020 [2]. Many sports disciplines use calisthenics exercises to train similar movements or to improve physical preparation in different sports [3,4]. The word calisthenics comes from the Greek “Kàlos” which means beauty and “Sthénos” strength. Today, with this term, we refer to a training technique that involves a series of free body exercises, performed with the aid of high bars, low bars, parallel bars, and rings. These exercises are often used by doctors, the military [5], and in schools [6,7] to improve individual physical performance.

Among the disciplines that make use of calisthenics exercises, we can find streetlifting. Streetlifting is an emerging strength discipline that blends the minimalist culture of calisthenics with the structured progression of weighted resistance training. While calisthenics has long emphasized control, body awareness, and relative strength, streetlifting introduces external loads to movements such as pull-ups and dips, allowing athletes to develop and measure absolute strength in a clear and competitive way. Notably, men’s streetlifting competitions include pull-ups on the bar, dips on the parallel bars, muscle-ups, and barbell squats, while women’s streetlifting programs through 2024 did not include ballasted muscle-ups. The rise in streetlifting is closely connected to the global calisthenics movement. Organizations such as the World Street Workout and Calisthenics Federation (WSWCF), founded in 2011, helped popularize bar-based training through international competitions and public events. In recent years, more specialized federations like the European Streetlifting Federation (ESLF) and national-level associations have begun formalizing the rules, weight classes, and judging standards for streetlifting competitions. These developments have strengthened the sport’s credibility and visibility [8,9,10].

Streetlifting is a sport strongly influenced by body weight and body composition. Competitions, in fact, involve a division into weight categories in order to allow athletes with similar body structure to compete [11]. Moreover, because performance depends heavily on strength-to-body weight ratio, nutritional strategies play a central role in supporting training adaptation, connective tissue, and health body composition (fat mass and muscle mass, hydration), which are all factors that can positively or negatively influence performance in this discipline.

From a physiological perspective, streetlifting involves high neuromuscular demand, emphasizing relative strength, scapulohumeral stability, tendon stiffness, and controlled force output under variable leverage. While research on calisthenics suggests that structured bodyweight training can improve muscular strength, endurance, and body composition [12], limited scientific literature specifically examines the metabolic, nutritional, and recovery demands of streetlifting athletes.

From what has just been cited, this narrative review aims to synthesize and combine current knowledge regarding body composition, nutrition, dietary supplementation, and sleep quality in streetlifting athletes, integrating evidence from related strength sports.

Sleep quality and nutrition parameters were included as they could play a central role in athlete well-being, body composition, and athletic performance.

### Study Design

This article takes a narrative review approach. This typology was chosen due to the scarcity of specific studies regarding nutrition and integration in streetlifting, which does not allow the realization of a systematic review or meta-analysis. Narrative revision instead offers the possibility of integrating evidence from similar strength disciplines, such as powerlifting and street workout. The search terms “streetlifting AND nutrition” (one result), “streetlifting AND body composition (one result)” and “streetlifting AND sleep quality” (zero results) on PubMed and Google Scholar were used for analysis. Given the limited number of publications related to streetlifting, studies conducted in other related strength sports (e.g., strength sport, calisthenics, powerlifting, street workout) have also been included. The inclusion criteria comprised peer-reviewed articles in English, reporting on nutrition, body composition, or sleep quality in strength disciplines. Non-peer-reviewed materials and studies unrelated to strength sports were not included. Searches were conducted from January 1996 to October 2025. In Table 1, we can find the various studies considered for this review.

## 2. Streetlifting and Body Composition

Streetlifitng and street workout are not synonymous but share some exercises and training methodology [32].

Street workout is characterized by accessibility and low cost, which would generate health benefits, such as reducing physical inactivity and obesity, and consequently helping to reduce cardiovascular risk factors and maintaining a healthy lifestyle [13]. Because performance in weighted pull-ups and dips is directly influenced by body mass, athletes often attempt to optimize body composition. This requires careful attention to caloric intake and macronutrient quality. Insufficient caloric intake may lead to reductions in lean mass or tendon resilience, increasing the risk of overuse injury. Conversely, excessive caloric surplus can lead to non-functional mass gain that negatively affects performance. The goal is to maintain lean mass stability and low-to-moderate body fat while ensuring sufficient energy availability for training and recovery.

The main somatotype of street workout athletes is balanced mesomorphic, which is similar to elite gymnasts [13]. The principal difference is that gymnasts have a greater mesomorphic component, and lower endomorphic and ectomoprhic values compared to street workout practitioners. The similarity in somatotype could be explained by the common biomechanical movements between these two sport disciplines [13].

Some research [13] proves that four months of street workout training is sufficient to shape the body. This finding is in line with the close relationship between morphology and function, where a balanced mesomorphic somatotype facilitates the street workout practice, working as a natural selector for its followers. In the future, cohort studies will be necessary, including pre- and post-measurements of a street workout program based on calisthenics to elucidate the influence of this discipline in morphology [13].

It is clear that practice of street workout plays the role of protecting against cardiovascular diseases, obesity, and mortality, due to the body composition, as well as the high muscle mass and low fat mass observed [14]. In fact, street workout could help to decrease body fat and blood pressure in young students [14] and also decrease overweight tendency in the similar way in the population [33]. A recent study showed that as the weight category increases, athletes are taller, having greater circumferences and endomorphic characteristics, which is in agreement with other studies conducted on competitive powerlifters [12,15,16,17]. Male streetlifting athletes included in the −66 kg and −74 kg categories have lower body fat percentages (7.5–7.7%), and great strength relative to their body weight in pull-ups, dips, and muscle-ups. In contrast, men in the +93 kg body weight category have a higher body fat percentage (14.2%) and lower strength-to-weight ratios in pull-ups, dips, and muscle ups. Similar trends were observed in the female gender [17].

According to studies, excess body fat can have a negative impact on prone traction performance in both sexes. Furthermore, both performances in supine tractions and in muscle-ups (performed only by men) present similar correlations with anthropometric and body composition parameters [18,19].

The squat represents the exercise included in streetlifting that has the greatest influence on the total score of the competition and unlike pull-ups, muscle ups, and dips, body fat may not be considered a limiting factor for performance [16]. This is confirmed by a study that found no relationship between fat mass and squatting (on a single lift) in elite male weightlifters [20].

Against these data [21], it can be said that squat performance was mainly supported by lean mass; furthermore, ectomorph biotype, leg height, and leg length showed an inverse correlation with squat performance, which is consistent with other studies [22], showing how tall athletes can be penalized by a greater balance shift than shorter athletes.

The importance of lean mass and the negative influence of fat mass suggest the adoption of training and nutritional strategies aimed at maximizing body composition [17] based on performance and therefore increasing the overall score of athletes in streetlifting competitions [17].

Furthermore, body composition takes into account the fact that BMI also plays a fundamental role in preventing injuries together with many other factors [23]. In fact, body composition is decisive in performance as seen from a study [17] on streetlifting which correlates different weight categories with different relative forces.

In fact, street workout also presents intrinsic risks related to injuries: physical activity levels [23], sport specialization [34], age [24], body mass index (BMI) [24], and sex [25]. The right way is a multifactorial approach to elucidate how factors are linked to injury risk and the need for nutritional intervention is highlighted to control weight and body composition in order to reduce the risk of injuries [24,26].

## 3. Nutritional Approach

To understand which nutritional approach is best suited to this discipline, it is necessary to study various aspects including calorie intake, protein intake, carbohydrate intake, and the use of supplements.

Streetlifitng is considered a strength sport as its objective is to seek maximum weight lifted for a single repetition in each of the four exercises mentioned previously.

It is known that our body can exploit three main energy metabolisms based on the intensity and duration of the physical effort made [35]. These three processes, different but closely intertwined, cooperate to meet the energy needs of the muscle. We therefore distinguish the anaerobic energy system, divided in turn into an alactacid component and a lactacid component, and the aerobic energy system [35].

Anaerobic pathways are able to provide ATP at high speed but for a limited time; in fact, they satisfy energy needs due to intense but short-term efforts, on the contrary, the aerobic system has the ability to generate large quantities of ATP but requires longer times, which is why it is aimed at efforts of lower intensity but longer duration [35].

However, the evidence shows that the energy is derived from each of the three metabolisms during almost all physical activities [35].

In particular, in streetlifting competitions, the duration of efforts is summarized in a handful of seconds in which the maximum intensity is expressed [23]. This can certainly translate into a large involvement of anaerobic metabolism linked both to the splitting of PCr for the production of ATP and to the anaerobic degradation of glucose through glycolysis and then lactic fermentation [35].

It therefore becomes clear that specific nutritional approaches are necessary to maximize performance in strength sports and therefore in streetlifting [35]. This is also reported by a study highlighting the importance of body composition on performance in streetlifting and the need to develop suitable nutritional strategies [17].

From a nutritional point of view, many studies have examined whether ingesting proteins and carbohydrates before or after resistance training can change their metabolism such that protein and glycogen balance is improved [27].

Accredited hypotheses are that an increased glycogen biosynthesis during recovery between sessions will improve the performance of subsequent workouts, and that an increase in protein synthesis and a reduction in protein degradation will lead to muscle hypertrophy and therefore an increase in lean body mass [27].

Furthermore, according to studies, men who were given whey protein (1.2–1.5 g/kg body mass) in addition to their diet had a greater increase in lean mass than the placebo group after 6 weeks of training with overloads [27,36].

From the results of these studies, it can be seen that an adequate diet in terms of energy (with a higher caloric intake than the daily requirement) in addition to a higher protein intake (1.2–1.5 g/kg of body weight) [37] promotes anabolism during training with overloads [27,38].

In conditions of energy deficit (energy introduced less than that consumed), however, there is some scientific evidence that a diet richer in proteins has advantages in preserving body muscle mass and strength [28].

Subsequently, much information has been explored with regard to the role of nutrition in influencing acute and chronic adaptations to training with overloads, on the quantities, quality, and optimal time to take in nutrients in order to obtain maximum benefits [38].

It has been seen that with the combination of exercises with overloads, an increase in amino acids and an increase in serum insulin appear to be additive in order to maximize the hypertrophic response. Furthermore, the time of taking the supplement is important to maximize the anabolic response. In fact, it has been seen that consuming a protein–glucid supplement close to exercise (for example, immediately before or immediately after exercise) sometimes seems to promote better recovery [38]. However, it is not clear whether and how these acute responses influence chronic adaptations [37].

Currently, the world of training is flooded with the supplement industry. Many products are sponsored and sold although some are not supported by scientific studies and have not been shown to significantly influence the acute response to training.

One of the few dietary supplements for which efficacy in improving muscle size and strength in combination with overload training has been repeatedly demonstrated as creatine [38].

The evidence proves that creatine supplementation can increase muscle strength by increasing intramuscular creatine stores and restoring ATP during and after intense muscle contractions [28]. Creatine greatly increases GLUT-4 content in the muscle membrane and, consequently, glucose uptake to enhance glycogen resynthesis [39,40]. The importance of suitable nutritional strategies to maximize body composition and streetlifting performance is also included in another study [17]

In conclusion, protein, carbohydrate ingestion time, and creatine supplementation can be important interventions to improve adaptations to training with overloads [38].

### Role of Sleep

Optimal recovery extends beyond nutrition to include sleep quality and overall stress management. A meta-analysis study showed that short-term sleep deprivation has a variable impact in many respects: simple attention, complex attention, working memory, processing speed, short-term memory, and reasoning [41]. The greatest effect was observed for simple and prolonged attention [29].

From a sporting point of view, optimal recovery extends beyond nutrition to include sleep quality and overall stress management. Adequate sleep supports muscle protein synthesis, hormonal regulation, motor learning, and central nervous system recovery [42].

One study [42], in particular, discusses the potential mechanisms linking sleep to body composition. Among the routes described, the possible role of growth hormone (GH) is highlighted to be involved in the homeostasis of all body tissues and strongly linked to brain activity that dominates deep sleep, which is also referred to as slow waves.

The importance of sleep for high performance varies based on the cognitive and physical needs of the sport itself. However, because most sports require the integration of more skills (aerobics, tactical skills, and technical skills), competitive athletes may benefit from increased sleep duration [43].

Furthermore, long-term sleep changes have been seen to be more likely to affect athletic performance than manipulations in the short term (deprivation or extension). Therefore, the impact of sleep on competitive athletes in achieving high performance depends on the demands of the sport. Currently, a limited number of studies investigating the quality and duration of sleep are available [43]. These are the reasons why we decided to include within this review the parameter of sleep quality as it is a determined factor for the health and general well-being of the athlete in addition to the hypothesized effect on performance.

This result would also appear to be associated with dietary models characterized by greater energy availability derived from proteins, and a lower energy share of carbohydrates and fats compared to those who sleep poorly [30].

Dietary proteins can influence sleep by the amino acid tryptophan (TRP) which promotes its initiation [44]. TRP is converted to 5-hydroxytryptophan (5-HTP) and then to serotonin, which is then metabolized to melatonin which serves as a sleep onset signal [45].

Research on sleep and nutrition is still in its infancy and some studies have suggested that meal timing and content may influence sleep [32].

## 4. Nutritional Education

A precise dietary approach can be supported by an increase in nutritional knowledge; in fact, a recent systematic review analyzed the effects of nutrition education interventions on eating habits, nutritional knowledge, body composition, and physical performance in athletes in team sports.

The main results showed that the implementation of nutritional education interventions consistently induced positive changes in eating habits and nutritional knowledge, as well as maintained or improved body composition [46,47].

Unfortunately, other studies show uneven results. This may be due to the wide range of intervention modalities and durations, limitations in methodologies, and the small number of case–control studies, making it difficult to draw firm conclusions about the effectiveness of interventions or which interventions were best [48].

These findings suggest that the health and performance of athletes in team sports from various backgrounds could benefit from nutrition education interventions as a complementary strategy to training routines [46]. Meanwhile, the disparity of results in a limited number of studies exploring the effectiveness of educational and nutritional interventions implies that more research on this topic is needed to draw definitive conclusions.

## 5. Limits of the Study

This review aims to increase knowledge in streetlifting, but given the scarcity of available data, it became necessary to integrate information from various studies, including those from different but similar disciplines, in terms of type of exercises, types of efforts performed, and equipment (Table 2). However, these studies are very heterogeneous with each other (reviews, RCT studies, studies on small samples, observational studies), making the results obtained less comparable.

## 6. Conclusions

Streetlifting is an emerging strength discipline that combines the neuromuscular demands of calisthenics with the progressive overload typical of weighted resistance training. Although its popularity and competitive structure continue to grow, the current scientific literature specifically addressing nutritional strategies, body composition management, and recovery protocols in streetlifting athletes remains limited.

Evidence derived from related strength sports shows that weight and body composition have an impact on streetlifting performance and help to prevent injuries.

In particular, the maintenance of low-to-moderate body fat levels, adequate energy availability, timing and quantity of protein, and carbohydrate and supplement intake promote lean mass accretion, tissue repair, and enhance adaptations to resistance exercises, which are also typical in streetlifting.

Other evidence cites the possible role of sleep in improving corporeal composition and performance, as well as the possible correlations of sleep with the nutritional aspect.

The data collected from various studies suggest the need to carry out studies that allow us to evaluate the impact of nutritional education on the variables associated with this discipline, in addition to precise treatments as already considered by other research.

Ultimately, we can state that this narrative revision appears to be the first to address and integrate more data in correlation with streetlifting, which to date cannot boast many studies in the literature. Given the growing interest and potential health and performance implications, future research should focus on controlled longitudinal studies that investigate specific nutritional needs, optimal macronutrient distribution, and recovery strategies tailored to streetlifting. Such evidence would contribute to developing standardized guidelines to support athlete safety and performance optimization, identifying the main weight categories of streetlifting competitions and researching the best nutritional strategies to obtain the maximum performance benefits based on the singularities and long-term health benefits. 

## Figures and Tables

**Table 1 nutrients-18-00105-t001:** Summary of the cited studies.

References	Population (No.)	Research Design	Aim of the Study	Outcome
[1]	untrained individuals (28)	controlled experimental intervention study	administer a calisthenics training protocol and evaluate its effects on posture, strength, and body composition	improved their posture, their strength, and their body composition
[2]	possible trends (38)	survey study	survey of fitness trends for 2020	calisthenics were identified as the seventh most popular activity trend in 2020
[3]	adult woman (35)	randomized controlled trial (RCT)	evaluated and compared the effectiveness of anaerobics–calisthenics and anaerobics/weight training on lower limb strength and body fat	improved selected strength and reduced body fat; nonsignificant differences between aerobic/calisthenics and aerobics/weight
[4]	male soccer players (18)	controlled intervention study	examine the effect of calisthenics exercises on body composition in soccer players	eight-week calisthenics exercise can provide positive development of the whole-body composition of soccer players
[5]	corps cadet (26)	controlled intervention study	determine the effects of high-intensity interval training (including calisthenics) in Corps cadets	HIT sustained fitness despite the short duration and reduced volume of activity
[6]	elite badminton players (321)	experimental training intervention study	investigate the effects of physiological functions of sprint interval training (SIT) on the aerobic capacity of elite badminton players	eight-week SIT effectively improved the aerobic exercise capacity of elite badminton players
[7]	healthy students at junior school (136)	intervention study	observe the physiological and psychological effects of Chinese traditional healthcare exercises (CTHEs) on the adolescents	junior school students would psychologically benefit from practicing CTHE
[11]	powerlifters (74)	cross-sectional observational study	provide normative performance and anthropometric data of Southern European classic powerlifters of both sexes, determine the possible relationships between these variables and performance, and develop population-specific predictive equations for single lifts and overall powerlifting performance	individuals with short arms excel in the bench press, individuals with shorter leg length excel in the squat, whereas individuals with longer arms relative to their height excel in the deadlift
[12]	healthy man moderately trained (23)	non-randomized trials	effect of progressive calisthenic push-up training on muscle strength and thickness	this study is the first to demonstrate that calisthenics, using different progressive variations to maintain strength training programming variables, can improve upper-body muscle strength
[13]	calisthenics athletes (14)	cross-sectional descriptive study	determines the characteristics and morphological profile of street workout athletes by means of anthropometry	balanced mesomorphic somatotype, low fat mass, and high muscle development, with dominance in the upper limbs and trunk
[14]	patients with CVD (6541)	cohort studies	evaluated the relation between components of body composition and mortality in patients with cardiovascular disease	BMI and lean mass help to decrease body fat and blood pressure
[15]	oceania competitive powerlifters (54)	cross-sectional correlational study	examined the anthropometric dimensions of powerlifters across various body mass (competitive bodyweight) categories	the powerlifters were highly mesomorphic and had large girths and bony breadths
[16]	semiprofessional rugby union players with some strongman training experience (23)	correlational observational study	to determine the relationships between anthropometric dimensions and maximal isoinertial strength to strongman performance	body structure and common gymnasium-based exercise strength are meaningfully related to strongman performance in novice strongman athletes
[17]	streetlifting athletes (79)	correlational study	relationships between anthropometric and strength profiles of streetlifting	the importance of lean mass and the negative influence of fat mass suggest the adoption of training and nutritional strategies
[18]	fit young men	non-randomized trials	the effect of experimental alterations in excess mass on pull-up performance	excess body fat can have a negative impact on prone traction performance
[19]	man (35) and woman (23)	correlational observational study	relationship of lat-pull repetitions and pull-ups	body composition components (lean body mass [LBM] and %fat) affected lat-pull max and pull-up max to a greater degree in men than in women
[20]	iranian elite weightlifters (42)	multivariate observational study	correlations of anthropometric and body composition variables with the performance of young elite weightlifters	strong correlations existing between weightlifter performance and the anthropometric and body composition variables
[21]	experienced resistance-trained men (20)	observational epidemiological study	the aim of this study was to investigate the relationships between muscle architecture, lower-body power, and maximal isometric force produced at midthigh pull	results show that force produced from the floor position may be more important than force produced from a position that mimics the second pull of the clean for deadlift and vertical jump performances
[22]	college-aged participants (21)	observational study	to investigate the relative contributions of biomechanical, anthropometric, and psychological variables to the prediction of maximum parallel barbell back squat strength	back squat strength in a heterogeneous population is multifactorial and more related to physical rather than psychological variables
[23]	community-dwelling women (909)	cohort studies	tracking of moderate-to-vigorous aerobic activity and resistance exercise behaviors and injuries	meeting versus not meeting guidelines were associated with more injuries during physical activity
[24]	male military (912)	cohort studies	injuries during sport	body composition (BMI) also plays a fundamental role in preventing injuries together with many other factors
[25]	all enlisted soldier accessions to the US Army between January 2011 and January 2014 (238 772)	retrospective cohort study	differences in the associations of body mass index, physical fitness, and tobacco use with lower extremity musculoskeletal injuries	among men, there were greater adjusted odds of injury among underweight (OR 1.27), overweight (OR 1.14), and obese soldiers (OR 1.54) (all *p* < 0.001) relative to normal-weight soldiers
[26]	practitioner (93)	case–control study	to investigate the characteristics and prevalence of injuries and variables that may increase these odds in street workout practitioners	shoulder and back (upper and mid) were the most injured body parts
[27]	males (36)	RCT study	to assess muscular adaptations during 6 weeks of resistance training	males who were supplemented with a combination of whey protein and creatine had greater increases in lean tissue mass and bench press than those who were supplemented with only whey protein or placebo
[28]	police officers (38)	non-randomized trials	compare the effects of a moderate hypocaloric, high-protein diet and resistance training, using two different protein supplements, versus hypocaloric diet alone on body compositional changes in overweight police officers	mean percent body fat with diet alone decreased from a baseline of 27 ± 1.8 to 25 ± 1.3% at 12 weeks. With diet, exercise, and casein, the decrease was from 26 ± 1.7 to 18 ± 1.1% and with diet, exercise, and whey protein the decrease was from 27 ± 1.6 to 23 ± 1.3%.
[29]	seventy articles	meta-analysis	to discover the effects of short-term SD on both speed and accuracy measures in six cognitive categories: simple attention, complex attention, working memory, processing speed, short-term memory, and reasoning	significant differences were observed for both speed and accuracy; however, there were no differences between speed and accuracy measures within each cognitive domain
[30]	participants (58)	randomized Controlled Trial	to assess the effect of protein intake during dietary energy restriction on indexes of sleep in overweight and obese adults	we showed that a higher protein quantity improved global sleep score independent of the protein source
[31]	patients with obstructive sleep apnea (19)	non-randomized trials	to investigate the role of a fatty meal before bedtime on sleep characteristics and blood pressure in patients with obstructive sleep apnea (OSA)	fatty meal intake before sleep can increase apnea index in OSA patients

**Table 2 nutrients-18-00105-t002:** Similarity and differences between sports.

Sport	Type	Muscle	Exercise
Calisthenics	Strength, flexibility, and coordination	Upper more than lower body	Skills and simple exercise
Gymnastics	Strength, flexibility, and coordination	Upper more than lower body	Skills and simple exercise
Street workout	Strength and endurance	Upper more than lower body	Exercise on the bar
Streetlifting	Strength	Upper and lower body	Muscle up, pull up, dip squat
Powerlifting	Strength	Upper and lower body	Bench press, deadlift, squat
Strongman	Strength	Upper and lower body	Different types of strength exercise
Weightlifting	Strength sport	Full-body sport	Accessory exercises

## Data Availability

No new data were created.

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
