# Peer review of "Performance Optimization in Streetlifting: The Combined Role of Nutrition, Lean Mass, and Sleep"

_nutrients, 2025, doi:10.3390/nu18010105_

Round 1
Reviewer 1 Report
Comments and Suggestions for Authors
Thank you for the opportunity to review your manuscript, “Performance Optimization in Streetlifting: The Combined Role of Nutrition, Lean Mass and Sleep”
The aim of this study was to summarize current knowledge regarding body composition, nutrition, supplementation, and sleep in streetlifting athletes by integrating findings from related strength sports.
The objective is described in the abstract, but it should be recalled and reformulated at the end of the introduction.
The manuscript is a narrative review focused on optimizing performance in streetlifting, an emerging strength sport based on calisthenics. It appears that streetlifting performance is closely linked to the strength-to-body-weight ratio, making it essential to maintain a low-to-moderate fat mass and adequate lean mass. From a nutritional perspective, evidence suggests that athletes must meet their energy requirements and consume a high protein intake, from 1.2 to 1.5 g/kg/day, to support muscle protein synthesis. Among supplements, creatine shows consistent benefits for maximal strength and ATP turnover. Regarding recovery, the duration and quality of sleep are fundamental for neuromuscular recovery, endocrine balance, and cognitive preparation.
Finally, the conclusion highlights that streetlifting athletes would benefit from integrated nutritional planning, the use of evidence-based supplements, and sleep optimization. Unfortunately, the lack of literature on streetlifting necessitates inferring and extrapolating performance, nutrition, and sleep strategies from other disciplines.
The manuscript would benefit from a summary table outlining the similarities and differences between the mentioned disciplines.
If the study's objective, as stated in the abstract, is to summarize current knowledge regarding body composition, nutrition, supplementation, and sleep in streetlifting athletes by integrating findings from related strength sports, the introduction should include a brief argument as to why the topic of nutrition and why the topic of sleep are being examined/explored.
Line 86-88: How many articles were found for each search term? . The authors should state the number of retrieved articles per search query.
Line 97 (and throughout): Please homogenize the spelling and nomenclature used for "street workout" (e.g., Line 109 uses "Street-Workout"; Line 112 uses "Street Workout").
Line 149-150: This statement must be referenced/supported by a citation.
A Limitations section should be included. This section should clearly reflect the scarcity of literature on Streetlifting, which obliges the manuscript to infer and extrapolate performance, nutrition, and sleep strategies from other disciplines.
The Conclusions section should not contain citations. All arguments must be supported and discussed in the body of the manuscript, and the conclusions should focus solely on summarizing the main findings.
Author Response
- The objective is described in the abstract, but it should be recalled and reformulated at the end of the introduction.
Thank you for the suggestion, we have added line 82-86
- The manuscript would benefit from a summary table outlining the similarities and differences between the mentioned disciplines.
As suggested, we have added the Table 2 in the Limits of the study section.
- The introduction should include a brief argument as to why the topic of nutrition and why the topic of sleep are being examined/explored.
We have argued at line 85-86, 172-173 and 260-263.
- Line 86-88: How many articles were found for each search term? The authors should state the number of retrieved articles per search query.
We have added at line 93-95 in round brackets.
- Line 97 (and throughout): Please homogenize the spelling and nomenclature used for "street workout" (e.g., Line 109 uses "Street-Workout"; Line 112 uses "Street Workout").
Sorry for the mistake, we have corrected at line 120.
- Line 149-150: This statement must be referenced/supported by a citation.
As suggested, we have added reference [25] line 161 and revised the reference section.
- A Limitations section should be included.
As requested, a Limits of the study section has been added at the end of the manuscript.
- The Conclusions section should not contain citations.
We thank the reviewer for raising this points that give to us the opportunity to revise the Conclusion section.
Reviewer 2 Report
Comments and Suggestions for Authors
This review aims to clarify the combined roles of nutrition, lean mass, and sleep in performance optimization in streetlifting. This review comprehensively examines the role of nutrition, body composition, and sleep in streetlifting. The reviewer believes that this review provides important information for the future development of streetlifting competitions. However, there are several limitations in this review.
- This review aims to clarify the combined roles of nutrition, lean mass, and sleep in performance optimization in streetlifting. The reviewer believes that maintaining and improving competitive results, physical performance, and physical condition is important not only in streetlifting but in all competitive sports. However, this review did not focus on competitive results, physical performance, or physical condition in streetlifting as primary outcomes. The reviewer considers that this review should be revised to focus on competitive results, physical performance, and physical condition in streetlifting as primary outcomes.
- This review describes the combined role of nutrition, lean mass, and sleep in performance optimization in streetlifting, but are the results of this review specific to streetlifting? The reviewer thinks that the authors should address whether the results presented in this review are specific to streetlifting.
- In relation to the above, the reviewer considers that nutrition and body composition play different roles in power-type competitive sports such as streetlifting than in other competitive sports. However, the reviewer believes that there is no difference in the role that sleep plays in power-type competitive sports and other competitive sports. Therefore, the reviewer does not understand the need to mention the role of sleep in this review. The authors should mention the importance of sleep in streetlifting compared to other competitive sports.
- In this review, the authors present the results of many previous studies on nutrition, body composition, and sleep in streetlifting. However, the previous studies presented in this review show results from previous studies with a variety of research designs, including observational studies and intervention studies etc. Therefore, the reviewer feels that the format and arguments of this review are not well organized because previous studies with various research designs are mixed. The reviewer recommens that previous studies be grouped by research design to give the review a proper appearance and organize the evidence.
- In this review, the authors present the results of many previous studies. However, the authors did not include any figures or tables in this review. The reviewer thinks that presenting the results of previous studies using figures and tables will make it easier for readers to understand.
Author Response
- The reviewer considers that this review should be revised to focus on competitive results, physical performance, and physical condition in streetlifting as primary outcomes.
We thank the reviewer for raising these points that give to us the opportunity to revised the manuscript, we have argued at line 73-75, 85-86, 161-163, 193-195, 233-235, 262-263.
2. The reviewer thinks that the authors should address whether the results presented in this review are specific to streetlifting.
As suggested also by the other reviewer, we have added a Limits of the study section and revised the Conclusion section.
3. The authors should mention the importance of sleep in streetlifting compared to other competitive sports.
Since there are no studies in the literature on the importance of sleep in streetlifting athletes (line 95), we had to refer to other sports not as a comparison but as similarities.
4/5. The reviewer recommens that previous studies be grouped by research design to give the review a proper appearance and organize the evidence. The reviewer thinks that presenting the results of previous studies using figures and tables will make it easier for readers to understand.
As requested, we have added the Table 1 in the Study design section and Table 2 in the Limits of the study section.
Round 2
Reviewer 1 Report
Comments and Suggestions for Authors
The authors have answered all my questions and made the corresponding changes to the manuscript.
Reviewer 2 Report
Comments and Suggestions for Authors
I think all responses to reviewers' comments have been addressed satisfactorily.
I have no comments on the revised manuscript.